# Gastrointestinal Colonization of Carbapenem-Resistant *Acinetobacter baumannii*: What Is the Implication for Infection Control?

**DOI:** 10.3390/antibiotics11101297

**Published:** 2022-09-22

**Authors:** Shuk-Ching Wong, Jonathan Hon-Kwan Chen, Pui-Hing Chau, Simon Yung-Chun So, Christine Ho-Yan AuYeung, Lithia Lai-Ha Yuen, Veronica Wing-Man Chan, Germaine Kit-Ming Lam, Kelvin Hei-Yeung Chiu, Pak-Leung Ho, Janice Yee-Chi Lo, Kwok-Yung Yuen, Vincent Chi-Chung Cheng

**Affiliations:** 1Infection Control Team, Queen Mary Hospital, Hong Kong West Cluster, Hong Kong SAR, China; 2Department of Microbiology, Queen Mary Hospital, Hong Kong SAR, China; 3School of Nursing, Li Ka Shing Faculty of Medicine, The University of Hong Kong, Hong Kong SAR, China; 4Department of Microbiology, Li Ka Shing Faculty of Medicine, The University of Hong Kong, Hong Kong SAR, China; 5Centre for Health Protection, Department of Health, Hong Kong SAR, China

**Keywords:** carbapenem-resistant *Acinetobacter baumannii*, multidrug-resistant *Acinetobacter baumannii*, gastrointestinal colonization, infection control

## Abstract

The epidemiology of patients with gastrointestinal colonization of carbapenem-resistant *Acinetobacter baumannii* (CRAB) has not been systematically analyzed. We aimed to analyze the incidence, risk factors, and clinical outcomes of patients with newly identified gastrointestinal colonization of CRAB in a healthcare region in Hong Kong, where a multi-pronged screening strategy for gastrointestinal colonization of CRAB, together with other multidrug-resistant organisms (MDROs), was conducted by collecting fecal specimens (rectal swab or stool) upon admission and during hospitalization. From 1 October 2015 to 31 December 2019, a total of 161,339 fecal specimens from 63,588 patients, 61,856 (97.3%) of whom were hospitalized patients, and 54,525 (88.1%) were screened upon admission, with 1309 positive for CRAB (2.4% prevalence). Among patients positive for CRAB in fecal specimens, 698 (53.3%) had newly detected gastrointestinal colonization of CRAB, giving an incidence of 10.03 per 10,000 patient admissions and constituting 2646 CRAB colonization days in the general wards. Excluding the 164 patients with co-colonization of other MDROs, 534 patients had gastrointestinal colonization with only CRAB, and 12.5% (67/534) developed symptomatic CRAB infections at a median of 61 days (range: 2 to 671 days), during prospective follow-up for 2 years. Compared with age- and sex-matched controls, patients being referred from residential care homes for the elderly, the presence of indwelling devices, use of beta-lactam/beta-lactamase inhibitors, carbapenems, and proton pump inhibitors in the preceding 6 months, and history of hospitalization in the past 6 months were significantly associated with gastrointestinal colonization with CRAB, as shown by multivariable analysis. Log-rank test showed that cases had significantly shorter survival duration than controls (*p* < 0.001). The adjusted hazard ratio of gastrointestinal colonization of CRAB was 1.8 (95% CI: 1.5–2.2; *p* < 0.001), as shown by Cox regression analysis. Whole-genome sequencing of eight patients with CRAB isolates in their blood cultures and rectal swabs during the same episode of hospitalization revealed ST-195 as the predominant type, as shown by multilocus sequencing type. Gastrointestinal colonization of CRAB poses a considerable challenge for infection prevention and control.

## 1. Introduction

*Acinetobacter baumannii* is one of the ESKAPE (*Enterococcus faecium*, *Staphylococcus aureus*, *Klebsiella pneumoniae*, *A. baumannii*, *Pseudomonas aeruginosa*, and *Enterobacter* species) pathogens which has excellent genome plasticity with high competence that enables it to take any genes from the surroundings and incorporate them into its genome. This results in acquisition of a wide repertoire of resistance genes, also called an antibacterial resistance gene island [1]. Carbapenem-resistant *A. baumannii* (CRAB) is an emerging Gram-negative organism primarily recognized in healthcare-associated infections. CRAB has been included as a critical pathogen on the priority list of antibiotic-resistant bacteria by the World Health Organization [2]. Carbapenem resistance rates among *A. baumannii* are particularly high in the hospitals of South and Southeast Asia, with an estimated prevalence of over 40% in most of the countries [3]. In China, CRAB strains were prevalent in 71% of the 77 ICUs recruited for nationwide surveillance. Clonal spread of CRAB was found in 38% of the ICUs surveyed, with a total of 22 different clones identified [4]. During the COVID-19 pandemic, the risk of CRAB co-infection or secondary infection increased among the patients hospitalized with COVID-19 [5]. The surge in CRAB infections could have been temporally related to the increasing COVID-19-related hospitalizations, resulting in changes to infection prevention and control practices. With the resumption of normal operation and infection control measures as COVID-19 hospitalizations decreased, cases of CRAB infections returned to the pre-COVID-19 level [6]. Control of CRAB infections in endemic hospitals remains a challenge. In addition to hand hygiene as a core component of infection prevention and control, active surveillance cultures, contact precautions, cohorting patients, environmental disinfection, use of chlorhexidine baths, and closing or stopping admissions to the ward have been adopted to eradicate CRAB infections from endemic hospitals, as illustrated in a recent review [7].

Active surveillance cultures for asymptomatic colonization of CRAB are not routine clinical practice, especially in resource-limited healthcare settings [8]. The use of active surveillance cultures, followed by isolation and enhanced contact precautions, is inversely associated with acquisition of CRAB and subsequent CRAB infection in the intensive care unit [9]. However, *A. baumannii* is found in multiple body sites with various rates of colonization [10]. There is no consensus on the choice of sampling sites for *A. baumannii* or on the use of a horizontal or vertical approach for screening during hospitalization [11]. Some studies suggested the skin has the highest yield in the detection of CRAB [12,13], whereas our previous study revealed fecal specimens to be positive in the highest number of patients with asymptomatic colonization of CRAB if a single sample is collected [14]. Fecal specimens, either rectal swab or stool, can be simultaneously screened for other multidrug-resistant organisms (MDROs), including carbapenem-resistant Enterobacterales (CRE), vancomycin-resistant Enterococci (VRE), and methicillin-resistant *S. aureus* (MRSA), in addition to CRAB and multidrug-resistant *A. baumannii* (MRAB), as proactive infection control measures [15,16,17,18,19].

In this study, we aimed to analyze the epidemiological characteristics of patients with newly identified gastrointestinal colonization of CRAB. The findings may have implications for the recommendation of infection prevention and control.

## 2. Materials and Methods

### 2.1. Setting

This was a retrospective study on the epidemiology of gastrointestinal colonization of CRAB in a healthcare network in Hong Kong West. It included an acute-care hospital (Queen Mary Hospital, a 1700-bed, university-affiliated teaching hospital providing a tertiary referral service) and five extended-care hospitals with another 1400 beds between 1 October 2015 and 31 December 2021.

### 2.2. Multi-Pronged Screening Strategy for Carbapenem-Resistant A. baumannii

A multi-pronged screening strategy for gastrointestinal colonization of CRAB, together with other MDROs, including MRAB, CRE, VRE, and MRSA, was conducted by collecting fecal specimens (rectal swab or stool) upon admission and during hospitalization, as previously described [15,16,17,18,19]. The analysis of gastrointestinal colonization of MRSA was reported previously [19]. The multi-pronged screening strategy comprised four major categories, targeted screening upon admission, opportunistic screening, safety net screening, and contact tracing. Briefly, targeted screening was performed in patients fulfilling the following criteria: (i) patients with a travel history of medical tourism or that had received an operation outside of Hong Kong in the past 12 months (an acronym of whom TO screen); (ii) patients with a history of hospitalization in Hong Kong in the past 3 months; (iii) patients admitted into special units such as the adult intensive care unit, renal dialysis center, hematology center, blood and marrow transplant center, liver transplant center, and heart-lung transplant center; and (iv) patients admitted into all medical wards. Opportunistic screening was performed for fecal specimens, which were sent to the microbiology laboratory for *Clostridioides difficile* culture or cytotoxin assay. Safety net screening was performed for all hospitalized patients at day 14. Contact tracing was conducted to identify the possible source or secondary cases if there was a patient with nosocomial acquisition of MDROs [20]. Apart from the multi-pronged screening strategy, universal admission screening was implemented in a specific clinical department for at least 3 months in case of persistent MDROs transmission. We selected the data of targeted screening upon admission for further analysis of gastrointestinal colonization of CRAB. 

### 2.3. Analysis of Gastrointestinal Colonization of Carbapenem-Resistant A. baumannii

For the purpose of epidemiological analysis, a new case of CRAB was defined if a patient had no preceding positive culture for CRAB in a clinical or screening specimen in the past 12 months. A new case of gastrointestinal colonization of CRAB was defined as a patient fulfilling the criteria of a new case of CRAB without isolation of CRAB in body sites other than in fecal specimens during the current episode of hospitalization. The epidemiology of patients with newly diagnosed gastrointestinal colonization of CRAB was described. The incidence of CRAB colonization in the gastrointestinal tract upon admission was compared with that of community-onset symptomatic CRAB infections as defined by the National Healthcare Safety Network of Centers for Disease Control and Prevention of the United States [21]. Each patient with newly identified gastrointestinal colonization of CRAB was prospectively followed up for 2 years (until 31 December 2021). The clinical outcome of developing symptomatic CRAB infections and the all-causes mortality were monitored. Infection control nurses assessed all patients with newly diagnosed CRAB at the bedside to determine if the case was classified as symptomatic CRAB infection or asymptomatic CRAB colonization in body sites other than the gastrointestinal tract. A case-control analysis was performed to identify the risk factors for gastrointestinal colonization of CRAB. Age- and sex-matched controls in a 1-to-2 ratio were selected within the same quarter as the diagnosis of a case. Controls were CRAB negative in the fecal specimens and had no positive culture of CRAB in a clinical or screening specimen in the past 12 months. Patients with co-colonization of MDROs in the gastrointestinal tract were excluded for the case-control analysis.

### 2.4. Data Source

The episode-based records of our patients were retrieved from the Clinical Data Analysis and Reporting System (CDARS), an electronic database of health records under the governance of the Hospital Authority, as previously described [22,23]. The quarterly number of patient days in our hospitals was also retrieved from CDARS. For all patients, a unique hospital number was assigned to each episode of admission. The duration of hospitalization for each admission was recorded. Episodes of each hospital admission involving a positive culture for *A. baumannii*, together with the results of antimicrobial susceptibility from the clinical and screening specimens were retrieved. 

### 2.5. Laboratory Identification of Carbapenem-Resistant A. bamannii

Fecal specimens (rectal swab or stool) collected from patients upon admission or during hospitalization were sent to the microbiology laboratory of Queen Mary Hospital. The rectal swab or around 1 g of stool was incubated in 2 mL of brain heart infusion enrichment broth with 10 μg vancomycin (Sigma-Aldrich, St. Louis, MO, USA) and 0.5 μg meropenem (Hospira, Melbourne, Australia) at 35 °C for 18 h. Ten microliters of the enriched broth was further subcultured onto MacConkey agar with 2 μg meropenem and incubated aerobically at 35 °C for 48 h, as previously described [17,24]. *A. baumannii* isolates were identified by matrix-assisted laser desorption/ionization time-of-flight mass spectrometry (Bruker Daltonics, Bremen, Germany). Antimicrobial susceptibility tests were performed using the Kirby–Bauer disk diffusion method according to the Clinical and Laboratory Standards Institute (CLSI) recommendations or manufacturer’s instructions. Antimicrobial susceptibility was defined according to the CLSI recommendations [25].

### 2.6. Whole-Genome Sequencing

Patients with fecal specimens and blood culture samples positive for CRAB with resistance to all antimicrobial agents during the same episode of hospitalization were subjected to whole-genome sequencing, as previously described [6,11]. DNA extraction was performed using Qiagen DNeasy Blood and Tissue Kit (Qiagen, Hilden, Germany) according to the manufacturer’s instructions. DNA libraries were prepared by the Nextera DNA Prep Kit (illumina Inc, San Diego, CA, USA) and the Nextera DNA CD Indexes (illumina Inc, CA, USA) according to the manufacturer’s instructions. The libraries were sequenced by loading onto the MiSeq sequencing system (illumina Inc, CA, USA) with 2 × 300 bp paired-end read run for 56 h. Prior to genome assembly, the quality of the raw sequencing reads was first evaluated using the FastQC and trimmed with Trimmomatic v0.39. De novo assembly on raw data reads was performed using Unicycler 0.4.9. Parsnp v. 1.7.4 was used for core-genome alignment and variant calls [26]. A maximum-likelihood phylogenetic tree based on total core-genome single-nucleotide polymorphisms (SNPs) was constructed by IQ-TREE 2 (http://www.iqtree.org, accessed on 5 September 2022) and viewed by FigTree v1.4.4 (http://tree.bio.ed.ac.uk/software/figtree, accessed on 5 September 2022). MLST was assigned by PubMLST (https://pubmlst.org/organisms/acinetobacter-baumannii, accessed on 5 September 2022) using the Oxford scheme.

### 2.7. Statistical Analysis

A logistic regression model with bivariate and multivariable analyses was used to determine the risk factors of gastrointestinal colonization of CRAB. In the bivariate analyses, one of the risk factors (i.e., referral from residential care home for the elderly, presence of indwelling devices, Charlson comorbidity index, use of beta-lactam/beta-lactamase inhibitors, cephalosporins, and fluoroquinolones in the preceding 6 months, and history of hospitalization in the past 6 months) was included in each model. In the multivariate analysis, all risk factors were selected to the model by stepwise variable selection. In addition, matching variables (i.e., age and sex) were included in the model, as suggested in the literature [27]. Kaplan-Meier survival analysis was conducted on the time to death from the diagnosis of gastrointestinal colonization of CRAB; patients who survived up to 31 December 2021 were considered as censored. Log-rank test was used to compare the statistical difference between the survival curves of the case and control groups. The effect of gastrointestinal colonization of CRAB on mortality was evaluated using the Cox regression model, adjusted for age and sex and other risk factors selected by stepwise variable selection. SPSS, version 27 (IBM), was used to perform the statistical analyses. All reported *p*-values were two-sided. A *p*-value of <0.05 was considered statistically significant.

## 3. Results

### 3.1. Multi-Pronged Screening Strategy for Carbapenem-Resistant A. baumannii

From 1 October 2015 to 31 December 2019, a total of 63,588 patients with 161,339 fecal specimens were screened in a healthcare network in Hong Kong West under the multi-pronged screening strategy for MDROs. Among the 63,588 patients, 61,856 (97.3%) were hospitalized patients, of which 54,525 (88.1%) were screened upon admission (Figure 1). The prevalence of CRAB was shown to be 2.4% (1309/54,525) by targeted screening upon admission. Among the 1309 CRAB-positive patients, 1063 (81.2%) were defined as new cases who had no culture positive for CRAB in clinical or screening specimens in the past 12 months. Out of these new CRAB cases, 746 (70.2%) were diagnosed in our acute-care hospital, and 698 (93.6%) fulfilled the definition of gastrointestinal colonization of CRAB. 

### 3.2. Analysis of Gastrointestinal Colonization of Carbapenem-Resistant A. baumannii

The overall incidence of gastrointestinal colonization of CRAB was 10.03 per 10,000 patient admissions (698 patients in 696,063 admissions), and 3.35 per 10,000 patient days (698 patients in 2,085,119 patient days), which was statistically higher than the incidence of community-onset symptomatic CRAB infections (6.39 per 10,000 patient admission and 2.13 per 10,000 patient days) during the study period (incidence rate ratio, 95% confidence interval (CI): 1.6, 1.5–1.7 and 1.6, 1.3–1.9; *p* < 0.001 for both rates).

Of the 698 patients with newly diagnosed gastrointestinal colonization of CRAB, 494 (70.8%) were admitted to the medical wards. In addition, 164 (23.5%) of the 698 patients had co-colonization of other MDROs, including MRSA (148 patients), CRE (10 patients), and *C. difficile* (six patients). Among the 148 patients with co-colonization of CRAB and MRSA in the gastrointestinal tract, a significantly higher proportion had indwelling devices (56.1%, 83/148 vs. 46.4%, 248/534; odds ratio (OR): 1.5; 95% CI: 1.0–2.1; *p* = 0.039) as compared to the 534 patients with gastrointestinal colonization of CRAB. 

For the remaining 534 patients with gastrointestinal colonization of CRAB only, 347 were male and 187 were female, with a median age of 85 years (range: 1 month to 103 years). Overall, 501 (93.8%) of the 534 patients were aged ≥61 years, including 46 (8.6%) aged 61–70 years, 90 (16.9%) aged 71–80 years, 240 (44.9%) aged 81–90 years, and 125 (23.4%) aged ≥91 years. Sixty-seven (12.5%) of the 534 patients developed symptomatic CRAB infections during prospective follow-up for 2 years. Two (3.0%) of the 67 patients had a positive blood culture, 29 (43.3%) had a positive culture in respiratory specimens, 22 (32.8%) in urine specimens, and 14 (20.9%) in wound specimens. Symptomatic infections occurred at a median of 61 days (range: 2 to 671 days) after the identification of gastrointestinal colonization of CRAB (Figure 2).

The epidemiological characteristics and risk factors for gastrointestinal colonization of *A. baumannii* are shown in Table 1. Briefly, patients referred from residential care homes for the elderly (RCHEs), presence of indwelling devices, and use of beta-lactam/beta-lactamase inhibitors, carbapenems, and proton pump inhibitors in the preceding 6 months, as well as history of hospitalization in the past 6 months, were significantly associated with gastrointestinal colonization of CRAB, as demonstrated in the multivariable analysis.

The all-causes mortality of patients with gastrointestinal colonization of CRAB was significantly higher than that of the controls in one year (56.7%, 303/534 vs. 25.9%, 277/1068; OR: 3.7; 95% CI: 3.0–4.7; *p* < 0.001) and in two years (69.7%, 372/534 vs. 35.1%, 375/1068; OR: 4.2; 95% CI: 3.4–5.3; *p* < 0.001). With Kaplan–Meier survival analysis, the median survival of cases and controls was 235 and 1232 days, respectively. Log-rank test showed that the cases had a significantly shorter survival duration than the controls (*p* < 0.001). The adjusted hazard ratio of gastrointestinal colonization of CRAB was 1.8 (95% CI: 1.5–2.2; *p* < 0.001) by Cox regression analysis.

### 3.3. Whole-Genome Sequencing

Eight patients had concomitant blood and rectal swab culture positive for CRAB isolates which demonstrated resistance to all antimicrobial agents, including ampicillin/sulbactam, piperacillin, piperacillin/tazobactam, ticarcillin/clavulanate, cefoperazone/sulbactam, ceftazidime, ciprofloxacin, gentamicin, amikacin, tobramycin, and cotrimoxazole, in addition to imipenem/cilastatin, during the same episode of hospitalization. The isolates were subjected to whole-genome sequencing. Of all eight patients, the isolates from the blood cultures and rectal swabs were clustered together per patient (Figure 3). Both isolates from the blood culture and rectal swab of patient 1 to patient 7 were all ST-195 multilocus sequence type (MLST), while both isolates from the blood culture and rectal swab of patient 8 were typed as ST-473 MLST.

## 4. Discussion

In this study, we demonstrated the incidence of gastrointestinal colonization of CRAB in a healthcare region over 4 years, involving more than 63,000 patients and 161,000 fecal specimens, which is the largest series in the literature. The prevalence of gastrointestinal colonization of CRAB was found to be 2.4% upon admission, which was lower than the corresponding findings of 4% to 12% in intensive care units, where the number of recruited patients was 63 to 565 [28,29,30]. The apparently higher prevalence of gastrointestinal colonization of CRAB upon admission to the intensive care unit could be related to the use of broad-spectrum antibiotics before admission to the intensive care units. In fact, the use of beta-lactam/beta-lactamase inhibitors, which included cefoperazone/sulbactam, ticarcillin/clavulanate, piperacillin/tazobactam, as well as carbapenems, in the preceding 6 months was significantly associated with gastrointestinal colonization of CRAB, as shown in our multivariable analysis involving 534 cases and 1068 age- and sex-matched controls. Use of carbapenems was also shown to be one of the significant risk factors for nosocomial colonization of CRAB detected in rectal swab specimens by multivariable analysis in 45 cases and 90 retrospective, matched controls [31].

The use of proton pump inhibitors (dexlansoprazole, esomeprazole, lansoprazole, omeprazole, pantoprazole, and rabeprazole) in the preceding 6 months was significantly associated with gastrointestinal colonization of CRAB. Gastric acid suppression was shown to increase the risk of gastrointestinal colonization of MDROs, including extended-spectrum beta-lactamase-producing Enterobacterales, carbapenemase-producing Enterobacterales, and VRE, by 75% in a primary meta-analysis of 12 studies, including 22,305 patients [32]. In addition to our previous finding of the association between proton pump inhibitors use and gastrointestinal colonization of MRSA [18], we also demonstrated the association between the use of proton pump inhibitors and gastrointestinal colonization of CRAB, which has not been reported in the literature and deserves further investigation.

The mechanisms of gastrointestinal colonization of *A. baumannii* are largely unknown. Secretory IgA was found to enhance the colonization of *A. baumannii* in the gastrointestinal tract, which was mediated by bacterial thioredoxin A in a mice model. A targeted deletion mutant of bacterial thioredoxin A reduced the bacterial load of *A. baumannii* in the gastrointestinal tract. In addition, the associated 50% lethal dose of a deletion mutant of bacterial thioredoxin A was increased nearly 100-fold in an intraperitoneal sepsis model, suggesting that bacterial thioredoxin A mediates gastrointestinal colonization as well as contributes to the pathogenesis of *A. baumannii* [33]. Further studies are warranted to investigate the mechanisms of gastrointestinal colonization of CRAB.

Gastrointestinal colonization of CRAB may have a significant impact on infection prevention and control. Patient-to-patient transmission of *A. baumannii* with gastrointestinal colonization was demonstrated in 17% of patients using multilocus sequence typing with subsequent pulsed-field gel electrophoresis for discrimination [34]. In the setting of the intensive care unit, patients with gastrointestinal colonization of *A. baumannii* had higher odds of bed rails contamination than those with only respiratory colonization [35]. Further study in the intensive care unit also revealed that 16% of environmental samples and 38% of air samples were positive in the rooms when the patients had gastrointestinal colonization of CRAB, whereas 10% of environmental samples and 13% of air samples were positive in the rooms when patients had respiratory colonization of CRAB [36]. Environmental contamination may serve as a persistent source leading to the outbreak of CRAB and MRAB [37,38,39,40,41,42], and most outbreaks occurred in intensive care units [37,38,40]. Air dispersal of MRAB also contributes to nosocomial transmission and outbreak [24]. 

The burden of gastrointestinal colonization of CRAB does not only affect intensive care units [28,29,30,34,37,38,40]. Of all 698 patients with newly diagnosed gastrointestinal colonization of CRAB, 71% were admitted to medical wards. In view of the limited number of isolation facilities, priority use of single-room isolation was given to patients with symptomatic CRAB infections instead of patients with asymptomatic gastrointestinal colonization of CRAB [23]. These 698 patients contributed to 2646 CRAB colonization days in the general wards, which may have posed a risk of nosocomial transmission. With the implementation of directly observed hand hygiene practice, we could achieve a decreasing trend of hospital-onset CRAB infections despite the burden of gastrointestinal colonization of CRAB in the general wards [23].

The challenge of gastrointestinal colonization of CRAB is increasing, as evidenced by the admission of patients colonized with CRAB from RCHEs. In our case-control analysis, a significantly higher proportion of patients with gastrointestinal colonization of CRAB was referred from RCHEs, which had the highest odds ratios among all factors, as shown in the multivariable analysis. In Hong Kong, RCHE is a collective term for all long-term care facilities and nursing homes that provide daily nursing care for their residents, who may require the use of feeding tubes, urinary catheters, and other medical devices [14,43,44]. The prevalence of MDROs, especially MRSA, was increasingly reported to be almost 50% in RCHEs during the COVID-19 pandemic [45]. In fact, CRAB also emerged in RCHEs with high background rates of MRSA. The prevalence of gastrointestinal colonization of CRAB was found to be 4.5% in 2015 [14], which was already higher than that in our current study. Therefore, infection prevention and control should be simultaneously implemented in acute-care and long-term care facilities, including RCHEs. In addition, a significantly higher proportion of patients with gastrointestinal colonization of CRAB had indwelling devices. The use of indwelling devices also correlates with gastrointestinal co-colonization of CRAB and MRSA, as shown in our study. Nursing care practice relating to indwelling devices should be revisited. Unrecognized CRAB transmission may have occurred in the hospitals because history of hospitalization in the past 6 months was also a significant risk factor for gastrointestinal colonization of CRAB.

The clinical impact of patients with gastrointestinal colonization of CRAB should be addressed. Similar to the patients with gastrointestinal colonization of MRSA, where symptomatic MRSA infections occurred in 14% of patients in a median of 112 days [19], symptomatic CRAB infections also developed in 13% of patients in a median of 61 days in this study. Decolonization of CRAB, as an important resistant Gram-negative organism, has received attention [46]. Gut microbiota modulation may play a role in gastrointestinal decolonization of MDROs, including CRAB, in the future [47].

Whole-genome sequencing was performed for the blood and fecal isolates of CRAB, which were resistant to all antimicrobial agents, collected from patients during the same episode of hospitalization. The clonality between the blood and fecal isolates was established in individual patients, which was similar to the finding of the whole-genome sequencing for the blood and fecal isolates of MRSA collected from patients during the same episode of hospitalization [19]. As shown in the whole-genome sequencing, ST-195 MLST was the predominant type, which was in contrast to our previously circulating strain of *A. baumannii* [48], suggesting that whole-genome sequencing should be regularly performed to update the molecular epidemiology of MDROs. In fact, whole-genome sequencing is increasingly used for molecular epidemiological analysis and outbreak investigations [24,49,50,51]. 

There were several limitations in this study. Firstly, we did not perform a full set of CRAB screenings by collecting specimens from multiple body sites. The yield of cultures from different body sites may not be comparable. The burden of asymptomatic colonization of CRAB was also underestimated. However, the comparison of cultures from different body sites was analyzed previously [12,13,14]. More importantly, we aimed to focus on the epidemiology of patients with gastrointestinal colonization of CRAB. Secondly, we did not demonstrate the change of incidence of gastrointestinal colonization of CRAB with respect to infection control measures, since our enhancement of infection control intervention using directly observed hand hygiene before meal and medication rounds for conscious, hospitalized patients was implemented before the commencement of this active surveillance culture program [23]. We also confined the study period to before the onset of the COVID-19 pandemic, where additional infection prevention and control was implemented which may have introduced confounding factors to our analysis [52,53,54,55,56,57].

## 5. Conclusions

Gastrointestinal colonization of CRAB is not uncommonly recognized upon admission, and its incidence is higher than community-onset symptomatic CRAB infections. Future studies are warranted to investigate the transmission dynamics, degree of environmental contamination, and potential regimen for gastrointestinal decolonization to minimize the risk of outbreaks in healthcare settings.

## Figures and Tables

**Figure 1 antibiotics-11-01297-f001:**
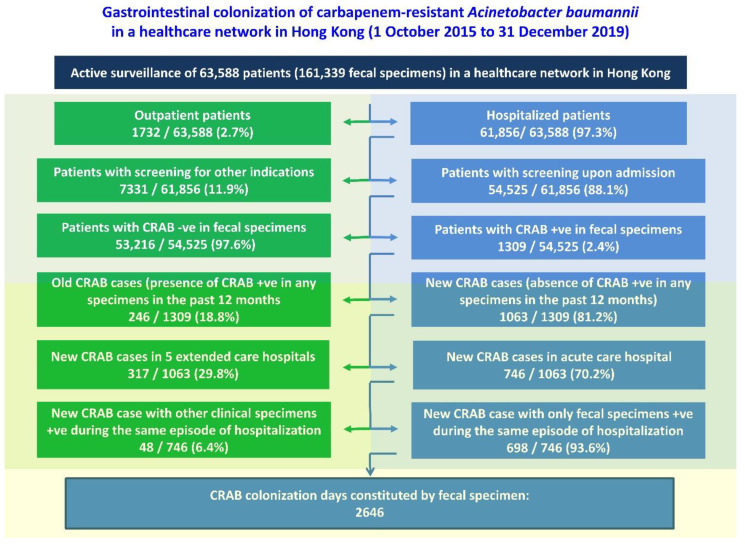
Gastrointestinal colonization of carbapenem-resistant *A. baumannii* (CRAB) in a healthcare network in Hong Kong.

**Figure 2 antibiotics-11-01297-f002:**
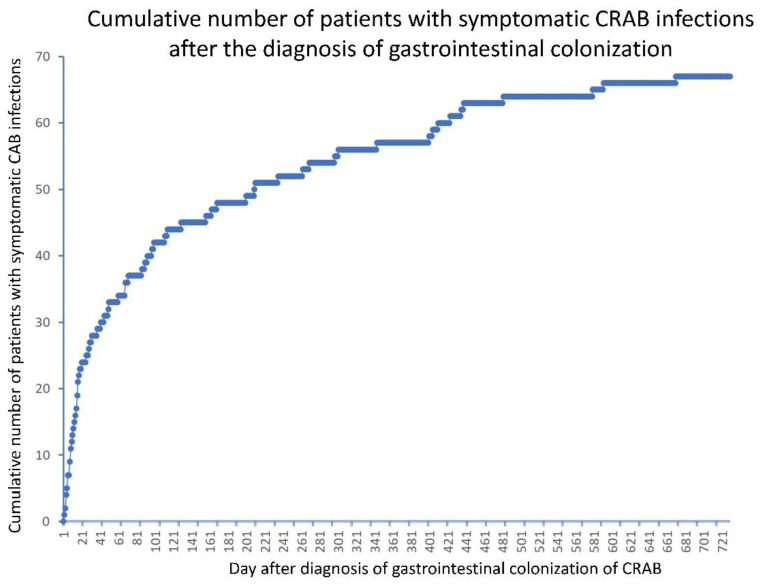
Patients with symptomatic carbapenem-resistant *A. baumannii* (CRAB) infections after the diagnosis of gastrointestinal colonization.

**Figure 3 antibiotics-11-01297-f003:**
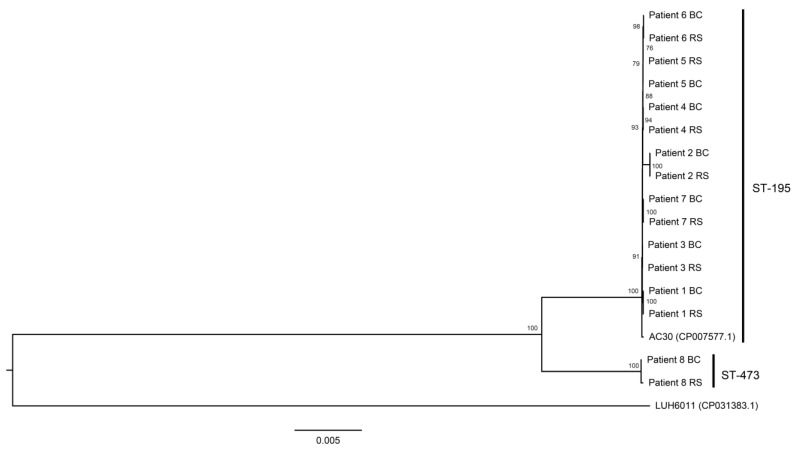
Maximum-likelihood phylogenetic tree of 16 CRAB isolates, which had resistance to all antimicrobial agents based on total core-genome single-nucleotide polymorphisms. These 16 CRAB isolates were also classified as MRAB and were collected from the blood culture (BC) and rectal swab (RS) from eight patients. Two MRAB control strains, LUH6011 (NCBI accession number CP031383.1) and AC30 (NCBI accession number: CP007577.1), were included as reference.

**Table 1 antibiotics-11-01297-t001:** Epidemiological characteristics for patients with or without gastrointestinal colonization of carbapenem-resistant *A. baumannii* (CRAB) in the age- and sex-matched controls ^a^.

Characteristics	Patients with GIC of CRAB (*n* = 534) ^b^	Patients without GIC of CRAB (*n* = 1068) ^c^	Bivariate Analysis ^d^		Multivariable Analysis ^d^	
			Odds Ratio (95% CI)	*p*-Value	Odds Ratio (95% CI)	*p*-Value
Patient referred from RCHE	373 (69.9%)	184 (17.2%)	11.1 (8.7–14.2)	<0.001	16.0 (11.6–22.0)	<0.001
Presence of indwelling device ^e^	248 (46.4%)	231 (21.6%)	3.1 (2.5–3.9)	<0.001	1.5 (1.1–2.1)	0.007
Charlson comorbidity index (mean ± SD)	4.3 ± 2.1	4.3 ± 2.3	1.0 (1.0–1.1)	0.876	NA ^f^	NA ^f^
Use of antibiotics in preceding 6 months ^g^						
Beta-lactam/beta-lactamase inhibitors	183 (34.3%)	86 (8.0%)	6.0 (4.5–7.9)	<0.001	2.3 (1.6–3.5)	<0.001
Cephalosporins	38 (7.1%)	27 (2.5%)	3.0 (1.8–4.9)	<0.001	NA ^f^	NA ^f^
Carbapenems	106 (19.9%)	29 (2.7%)	8.9 (5.8–13.6)	<0.001	4.2 (2.5–7.3)	<0.001
Fluoroquinolones	69 (12.9%)	56 (5.2%)	2.7 (1.9–3.9)	<0.001	NA ^f^	NA ^f^
Use of PPI in preceding 6 months ^h^	296 (55.4%)	247 (23.1%)	4.1 (3.3–5.2)	0.001	1.7 (1.3–2.4)	<0.001
Hospitalization in the past 6 months	456 (85.4%)	515 (48.2%)	6.3 (4.8–8.2)	<0.001	3.5 (2.4–4.9)	<0.001

CI, confident interval; GIC, gastrointestinal colonization; NA, not applicable; PPI, proton pump inhibitors; RCHE, residential care home for the elderly; SD, standard deviation. ^a^ Patients with co-colonization of carbapenem-resistant Enterobacterales, methicillin-resistant *S. aureus*, vancomycin-resistant Enterococci, and *C. difficile* were excluded for case-control analysis. ^b^ Case patients had isolation of CRAB in fecal specimen and had no isolation of CRAB in the other clinical or screening specimens during hospitalization. ^c^ Age- and sex-matched patients staying in the same clinical specialty within the same quarter of a CRAB case and with negative result of CRAB in fecal and clinical or screening specimens were chosen as controls in a ratio of 1:2. ^d^ An odds ratio significantly greater than 1 implied higher risk of having gastrointestinal colonization of CRAB. In the multivariable model, risk factors were selected by stepwise variable selection, and age and sex were entered. ^e^ Indwelling devices included nasogastric tube, tracheostomy or endotracheal tube, urinary catheter, and drain. ^f^ Variables were not selected by the stepwise procedure. ^g^ Beta-lactam/beta-lactamase inhibitors included cefoperazone/sulbactam, ticarcillin/clavulanate, and piperacillin/tazobactam; cephalosporins included cefotaxime, ceftriaxone, ceftazidime, cefepime, and ceftaroline; carbapenems included meropenem, imipenem/cilastatin, and ertapenem; fluoroquinolones included ciprofloxacin, levofloxacin, and moxifloxacin. ^h^ Proton pump inhibitors included dexlansoprazole, esomeprazole, lansoprazole, omeprazole, pantoprazole, and rabeprazole.

## Data Availability

Data are available upon reasonable request.

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
