# Peer review of "Gastrointestinal Colonization of Carbapenem-Resistant Acinetobacter baumannii: What Is the Implication for Infection Control?"

_antibiotics, 2022, doi:10.3390/antibiotics11101297_

Round 1

Reviewer 1 Report

This manuscript fitted well in the scope of journals recent issue on antimicrobial resistance and infection prevention and control. The timely update of AMR in bacterial pathogen especially on ESKAPE pathogens is necessary and important. Authors nicely addressed increasing incidences of gastrointestinal colonization of CRAB in a hospitalized subjects. Authors retrospectively evaluated patient data of 4 years, involving 60K patients is strength  of this manuscript. However, there are some limitations and corrections need to be addressed before its publication. Here are my overall comments.

Major comments

1.  Introduction parts need to improved, A. baumannii is one of the ESKAPE pathogen which has excellent genome plasticity with high competence that enable it takes any genes from surroundings and incorporate into its genome. It resulted into a acquisition of wide repertoire of resistance genes, which also called a antibacterial resistance gene island. please include this part in the introduction.

2. Authors showed the isolation of CRAB (1309/54525, 2.4%), how many A. baumannii isolates were isolated in this study, please mention did all the remaining n=54525 isolates were A. baumannii. What are criteria for inclusion, which methods used for confirmation. Please define the criteria for CRAB, what MIC values of carbapenem were taken for the confirmation of CRAB. 

3. Discussion part line 289, about BL-BLI combination, please mention the name of BL-BLI combinations used, will be easy for the readers to understand the statement. line 295 please do the same for proton pump inhibitor.

4. Patient consent and Ethical statement is missing.  

Minor Comments

Line 46: Please change "agent" to "pathogen".

Line 67: Please change "Acinetobacter baumannii" to A. baumannii, here and here after everywhere. Please do the same for other bacteria as well. 

Line 215: What is CPE? Please define or did you mean CRE please check.

Author Response

Reply to Reviewer 1

Open Review

This manuscript fitted well in the scope of journals recent issue on antimicrobial resistance and infection prevention and control. The timely update of AMR in bacterial pathogen especially on ESKAPE pathogens is necessary and important. Authors nicely addressed increasing incidences of gastrointestinal colonization of CRAB in a hospitalized subjects. Authors retrospectively evaluated patient data of 4 years, involving 60K patients is strength  of this manuscript. However, there are some limitations and corrections need to be addressed before its publication. Here are my overall comments.

Major comments

  1. Introduction parts need to improved, A. baumannii is one of the ESKAPE pathogen which has excellent genome plasticity with high competence that enable it takes any genes from surroundings and incorporate into its genome. It resulted into a acquisition of wide repertoire of resistance genes, which also called a antibacterial resistance gene island. please include this part in the introduction.

Ans: Thank you very much for your suggestion.

We have included this sentence in the beginning of Introduction.

Acinetobacter baumannii is one of the ESKAPE (Enterococcus faecium, Staphylococcus aureus, Klebsiella pneumoniae, Acinetobacter baumannii, Pseudomonas aeruginosa, and Enterobacter species) pathogens which has excellent genome plasticity with high competence that enables it to take any genes from the surroundings and incorporate into its genome. This results in acquisition of wide repertoire of resistance genes, also called an antibacterial resistance gene island [reference as follows].

De Oliveira DMP, Forde BM, Kidd TJ, Harris PNA, Schembri MA, Beatson SA, Paterson DL, Walker MJ. Antimicrobial Resistance in ESKAPE Pathogens. Clin Microbiol Rev. 2020 May 13;33(3):e00181-19. doi: 10.1128/CMR.00181-19. PMID: 32404435; PMCID: PMC7227449.

  1. Authors showed the isolation of CRAB (1309/54525, 2.4%), how many A. baumannii isolates were isolated in this study, please mention did all the remaining n=54525 isolates were A. baumannii. What are criteria for inclusion, which methods used for confirmation. Please define the criteria for CRAB, what MIC values of carbapenem were taken for the confirmation of CRAB. 

Ans: Thank you very much for your question.

“The isolation of CRAB (1309/54525, 2.4%)” referred to the gastrointestinal colonization of CRAB in 1309 patients out of 54,525 patients who were subjected to admission screening. It did not mean the presence of 54,525 bacterial isolates of A. baumannii.

In our laboratory protocol, we added the fecal specimens (rectal swabs or stool) into enrichment broth with antimicrobial agents including meropenem. Organisms growing in the enrichment broth were resistant to meropenem. The organisms were identified by matrix-assisted laser desorption/ionization time-of flight mass spectrometry. Therefore, we can only detect carbapenem-resistant Acinetobacter baumannii in our laboratory protocol, but not the number of patients with gastrointestinal colonization of carbapenem-susceptible Acinetobacter baumannii.

The criteria for CRAB is based on antimicrobial susceptibility tests using the Kirby-Bauer disk diffusion method according to the Clinical and Laboratory Standards Institute. We did not obtain MIC values of carbapenems.

The information is presented in section “2.5. Laboratory Identification of Carbapenem-resistant A. bamannii” in the manuscript.

“The rectal swab or around 1 g of stool were incubated in 2 ml of brain heart infusion enrichment broth with 10 μg vancomycin (Sigma-Aldrich, St. Louis, MO, USA) and 0.5 μg meropenem (Hospira, Melbourne, Australia) at 35 °C for 18 h. Ten micro-litters of the enriched broth was further subcultured onto MacConkey agar with 2 μg meropenem and incubated aerobically at 35 °C for 48 h as previously described. A. baumannii isolates were identified by matrix-assisted laser desorption/ionization time-of flight mass spectrometry (Bruker Daltonics, Bremen, Germany). Antimicrobial susceptibility tests were performed using the Kirby-Bauer disk diffusion method according to the Clinical and Laboratory Standards Institute (CLSI) recommendations, or manufacturer’s instructions. Antimicrobial susceptibility was defined according to the CLSI recommendations.”

  1. Discussion part line 289, about BL-BLI combination, please mention the name of BL-BLI combinations used, will be easy for the readers to understand the statement. line 295 please do the same for proton pump inhibitor.

Ans: Thank you very much for your suggestion.

Beta-lactam/beta-lactamase inhibitors included cefoperazone/sulbactam, ticarcillin/clavulanate, piperacillin/tazobactam.

Proton pump inhibitors included dexlansoprazole, esomeprazole, lansoprazole, omeprazole, pantoprazole, and rabeprazole.

We have added these names in the revised discussion.

  1. Patient consent and Ethical statement is missing.  

Ans: Thank you very much for your suggestion.

The Ethical statement and Patient consent are included at the end of this manuscript according to the journal format.

Institutional Review Board Statement: The study was conducted in accordance with the Declaration of Helsinki and approved by the Institutional Review Board of the University of Hong Kong/Hospital Authority Hong Kong West Cluster (UW 22-591) on 25 August 2022.

Informed Consent Statement: Informed consent was not required because the data was retrospectively obtained from an anonymous microbiology database.

Minor Comments

Line 46: Please change "agent" to "pathogen".

Ans: Thank you very much for your suggestion. We have changed "agent" to "pathogen" accordingly.

Line 67: Please change "Acinetobacter baumannii" to A. baumannii, here and here after everywhere. Please do the same for other bacteria as well. 

Ans: Thank you very much for your suggestion. We have changed "Acinetobacter baumannii" to “A. baumannii”.

In addition, we also changed “Clostridioides difficile” to “C. difficile”, “Staphylococcus aureus” to “S. aureus” when the names appeared again in the the revised manuscript. 

Line 215: What is CPE? Please define or did you mean CRE please check.

Ans: Thank you very much for your question. We have corrected the typo and changed CPE to CRE in the revised manuscript.

Reviewer 2 Report

1. Since the gut microbiome is considered to be varied according to age, at what range it showed its impact on gastrointestinal colonization of CRAB?

2. In line no 305, instead of the secretory IgA Secretary word used, please look for any such spelling mistakes.

3. Explain why targeted screening upon admission was selected apart from the others?

4. Does the screening method show any variation in the results?

Author Response

Reply to Reviewer 2

Open Review

  1. Since the gut microbiome is considered to be varied according to age, at what range it showed its impact on gastrointestinal colonization of CRAB?

Ans: Thank you very much for your question. Of 534 patients with gastrointestinal colonization of CRAB only, the median age was 85 years (range: 1 month to 103 years). Further analysis showed that 93.8% (501/534) patients were aged ≥ 61 years, including 46 (8.6%) aged 61-70 years, 90 (16.9%) aged 71-80 years, 240 (44.9%) aged 81-90 years, and 125 (23.4%) aged ≥ 91 years.

We have revised the result section as below:

Overall, 501 (93.8%) of 534 patients were aged ≥ 61 years, including 46 (8.6%) aged 61-70 years, 90 (16.9%) aged 71-80 years, 240 (44.9%) aged 81-90 years, and 125 (23.4%) aged ≥ 91 years.

  1. In line no 305, instead of the secretory IgA Secretary word used, please look for any such spelling mistakes.

Ans: Thank you very much for your question. We have corrected the typo and changed “Secretary IgA” to “Secretory IgA” in the revised manuscript.

  1. Explain why targeted screening upon admission was selected apart from the others?

Ans: Thank you very much for your question.

Our multi-pronged screening strategy comprised four major categories, which included targeted screening upon admission, opportunistic screening, safety net screening, and contact tracing.

Targeted screening upon admission was selected for analysis because we could calculate the incidence of gastrointestinal colonization of CRAB per 10,000 patient admissions, which was found to be 10.03 per 10,000 patient admissions (698 patients in 696,063 admissions).

In the revised manuscript, we added a sentence as below:

 “We selected the data of targeted screening upon admission for further analysis of gas-trointestinal colonization of CRAB.” in the section “2.2. Multi-pronged Screening Strategy for Carbapenem-resistant A. baumannii” 

  1. Does the screening method show any variation in the results?

Ans: Thank you very much for your question. As we only selected the data of targeted screening upon admission for further analysis of gastrointestinal colonization of CRAB, we did not have the data of gastrointestinal colonization of CRAB in the other screening strategies (opportunistic screening, safety net screening, and contact tracing).

Reviewer 3 Report

The manuscript "Gastrointestinal Colonization of Carbapenem-resistant Acinetobacter baumannii: What is the Implication on Infection Control" addresses incidence, risk factors, and clinical outcomes of patients with newly identified gastrointestinal colonization of carbapenem-resistant Acinetobacter baumannii. Patients were screened for CRAB and other multidrug-resistant organisms and whole genome sequencing was also applyied to investigate the predominant type of CRAB.

Prior to publication, it will be necessary to perform only small adjustments:

1) Increase the size of Figure 3. It is very difficult to read the information presented in the pdf file.

2) Present an aditional paragraph at the end of discussion addressing future prospects and perspectives.

3) Perform minor spell check to remove typos. 

Author Response

Reply to Reviewer 3

Open Review

The manuscript "Gastrointestinal Colonization of Carbapenem-resistant Acinetobacter baumannii: What is the Implication on Infection Control" addresses incidence, risk factors, and clinical outcomes of patients with newly identified gastrointestinal colonization of carbapenem-resistant Acinetobacter baumannii. Patients were screened for CRAB and other multidrug-resistant organisms and whole genome sequencing was also applied to investigate the predominant type of CRAB.

Prior to publication, it will be necessary to perform only small adjustments:

1) Increase the size of Figure 3. It is very difficult to read the information presented in the pdf file.

Ans: Thank you for your suggestion. The font size of Figure 3 has been increased in the revised manuscript.

2) Present an additional paragraph at the end of discussion addressing future prospects and perspectives.

Ans: Thank you for your suggestion. We have added an additional paragraph as “Conclusion” to address the future prospects and perspectives as below.

Gastrointestinal colonization of CRAB is not uncommonly recognized upon admission, and its incidence is higher than community-onset symptomatic CRAB infections. Future studies are warranted to investigate the transmission dynamics, degree of environmental contamination, and potential regimen for gastrointestinal decolonization to minimize the risk of outbreaks in healthcare settings.

3) Perform minor spell check to remove typos. 

Ans: Thank you for your suggestion. We have performed spell check to remove typos accordingly.

Reviewer 4 Report

In the article by Wong et al., the authors surveyed the incidence, risk factors, and clinical outcomes of gastrointestinal carbapenem-resistant Acinetobacter baumannii (CRAB) infection in patients. They used fecal samples from patients to understand the factors associated with the gastrointestinal colonization of CRAB. Overall, the study is well-conducted, and conclusions supported by the data. The study will be of interest to readers in the current scenario of steadily increasing drug resistance amongst various pathogens, including CRAB which is difficult to treat and has been enlisted amongst the priority list of antibiotic-resistant bacteria that are of global concern by WHO.   

Minor comments:

1.     Fig 1 – The statement “unrecognized CRAB colonization …. specimen” is not very clear. It will be better to clarify what the authors meant.

2.     Fig 3 - The labeling is illegible. Consider magnifying with larger fonts.

Author Response

Reply to Reviewer 4

In the article by Wong et al., the authors surveyed the incidence, risk factors, and clinical outcomes of gastrointestinal carbapenem-resistant Acinetobacter baumannii (CRAB) infection in patients. They used fecal samples from patients to understand the factors associated with the gastrointestinal colonization of CRAB. Overall, the study is well-conducted, and conclusions supported by the data. The study will be of interest to readers in the current scenario of steadily increasing drug resistance amongst various pathogens, including CRAB which is difficult to treat and has been enlisted amongst the priority list of antibiotic-resistant bacteria that are of global concern by WHO.   

Minor comments:

  1. Fig 1 – The statement “unrecognized CRAB colonization …. specimen” is not very clear. It will be better to clarify what the authors meant.

      Ans: Thank you for your question. Our intention to use the word “unrecognized” is to highlight the importance of our active surveillance program. Without this screening program, these patients with gastrointestinal colonization of CRAB, without concomitant colonization of CRAB in other body sites, would be missed.

      To avoid the misunderstanding, we have removed the term “unrecognized” in the revised manuscript.

  1. Fig 3 - The labelling is illegible. Consider magnifying with larger fonts.

Ans: Thank you for your suggestion. The font size of Figure 3 has been increased in the revised manuscript.

Round 2

Reviewer 2 Report

All the queries were answered.